# OpenReview forum: "RFEval: Benchmarking Reasoning Faithfulness under Counterfactual Reasoning Intervention in Large Reasoning Models"
_ICLR.cc/2026/Conference — ICLR 2026 Poster_

### Official Review · Reviewer_WEsV · 2025-10-29

**Soundness:** 3
**Presentation:** 3
**Contribution:** 3
**Rating:** 6
**Confidence:** 3

**Summary:**

This paper defines reasoning faithfulness in terms of stance consistency and causal influence, independent of accuracy. It introduces RFEval, a 7,186-instance benchmark across seven reasoning tasks that uses counterfactuals to test model responses. Evaluation of 12 open-source models shows unfaithfulness in about half of the cases, especially due to stance inconsistency. Math and code tasks are most challenging, and training methods impact faithfulness more than model size. Accuracy alone is not a reliable indicator of faithfulness.

**Strengths:**

1. An interesting benchmark for evaluating the faithfulness of LRMs and a clear formalization of reasoning faithfulness (stance consistency and causal influence) are presented in the paper.
2. This paper constructs a broad, heterogeneous benchmark (7 tasks, 7,186 items), enabling cross-family, cross-task patterns.

**Weaknesses:**

1. The definition and computation of Stance Consistency seem misaligned with how large reasoning models (LRMs) behave on complex tasks. In a genuine reasoning process, each step should not be assumed to be consistently aligned with the last step. For example, as mentioned in DeepSeek-R1, LRMs often experience "aha moments," where they explore different reasoning paths and self-correct after encountering errors. This is generally regarded as a normal behavior for LRMs on complex problems. However, the notion of Stance Consistency as used in this paper appears more suitable for low-difficulty tasks where reasoning is simple and linear, with little trial-and-error or backtracking. This is exemplified by the use of GSM8K in the Math portion, where the problems are relatively straightforward and the reasoning path is direct.
2. Results emphasize that hybrid SFT+RL pipelines correlate with higher RF, but comparisons span different families, data mixtures, and training details. Without controlled ablations (same base model, held-out data), causal interpretations about the training recipe remain tentative and unreliable.
3. The scope is limited to open-source LRMs. Although the authors argue that evaluating closed-source models is challenging, this limitation represents a notable weakness of the benchmark and appears to stem from its design choices.

**Questions:**

1. Since the externally injected reasoning is not generated by the model itself, the evaluation reflects whether the model adheres to the external reasoning rather than to its own reasoning process. This mismatch raises concerns about the validity of the conclusions drawn. How significant is the impact of this mismatch? For instance, what would the results look like if we evaluate reasoning faithfulness directly on the original model outputs (without considering the injected reasoning r') rather than on the counterfactual contrast?

---

> ### Author Response · Authors · 2025-11-26
>
> Thank you for your thoughtful and detailed review. Below, we respond to each of your points in turn.
>
> ### Weakness 1: Clarification of Stance Consistency (Eq. 2)
>
> ---
>
> > **Reviewer Concern**: The definition and computation of Stance Consistency seem misaligned with how LRMs behave on complex tasks. In a genuine reasoning process, each step should not be assumed to be consistently aligned with the last step (e.g., "aha moments" in DeepSeek-R1). The notion of Stance Consistency appears more suitable for low-difficulty tasks where reasoning is simple and linear, with little trial-and-error or backtracking.
> >
>
> > **Summary**: We also have evaluated LRMs on **a broad tasks that includes complex, non-linear domains**, where we indeed **observe exploratory, backtracking, “aha-moment” behavior**. Our stance-consistency metric treats **explicit self-corrections as faithful**, but counts **silent internal backtracking as inconsistent**, since unmarked reversals are indistinguishable from incoherent reasoning for end users.
> >
>
> Thank you for pointing out the uniqueness of LRM. We agree that exploration, backtracking, and “aha moments” are characteristic of LRMs on complex tasks and should not automatically be treated as failures.
>
> Importantly, our evaluation is not restricted to simple, linear problems like GSM8K. RFEval **includes more complex and non-linear domains** such as LiveCodeBench (code generation), MMLU (including law), and other tasks. In these settings, **we do observe exactly the kind of exploratory and self-corrective behavior** the reviewer describes. For example, the legal-decision case in Fig. 18 shows a model initially following an incorrect line of reasoning, then explicitly questioning and retracting it (“**But wait—is that always true? Actually, no!**”) before converging to the correct stance. **Our metric classifies this as stance-consistent** and faithful because $\text{Identified}(u,v)=1$ for the correction step.
>
> In the revision, we will more explicitly clarify that Stance Consistency is defined at the level of justified stance changes, not as a prohibition on backtracking. A limitation of our design is that **purely “silent” internal backtracking**—where the model reverses stance without acknowledging the earlier error—**will be treated as inconsistent**. This is intentional: from a user-facing faithfulness perspective, unmarked reversals are indistinguishable from incoherent reasoning, whereas **explicit identification of the flaw is what makes an “aha moment” transparent and trustworthy**.

---

> ### Author Response · Authors · 2025-11-26
>
> ### Weakness 2: Ablation of A3
>
> ---
>
> > **Reviewer Concern**: Results emphasize that hybrid SFT+RL pipelines correlate with higher RF, but comparisons span different families, data mixtures, and training details. Without controlled ablations (same base model, held-out data), causal interpretations about the training recipe remain tentative and unreliable.
> >
>
> > **Summary**: We supplement the cross-family analysis with **within-family ablations** and a **reward analysis**, showing that **RL-style post-training and reward design can be associated with changes in $\text{RF}$ and do not explicitly reward stance consistency or causal influence**. Accordingly, we now present the post-training regime as an important observational correlate of $\text{RF}$ rather than a fully deconfounded causal driver.
> >
>
> Thank you for raising this point. We agree that cross-family comparisons in Table 2 alone cannot support strong causal claims about training recipes, since architectures and data mixtures differ across model families. To address this, we supplement **the cross-family view with within-family ablations** (MiMo-7B, Olmo-3-7B) and **a reward analysis** on the DeepSeek-R1 family. These indicate that the choice of RL-style post-training and reward design can be associated with changes in $\text{RF}$ and does not explicitly reward stance consistency or causal influence. For the detailed experimental setup and numerical results, we kindly ask you to refer to our 2nd response to Reviewer JhME.
>
> In the revised paper (A3 in Sec. 5), we therefore frame post-training regime as an important observational correlate of $\text{RF}$ rather than a fully deconfounded causal driver.

---

> ### Author Response · Authors · 2025-11-26
>
> ### Weakness 3: Limited Scope of Evaluation
>
> ---
>
> > **Reviewer Concern**
> The scope is limited to open-source LRMs. Although the authors argue that evaluating closed-source models is challenging, this limitation represents a notable weakness of the benchmark and appears to stem from its design choices.
> >
>
> > **Summary**
> To provide signal beyond open-source models, we ran a **small multi-turn study on SOTA closed-source LRMs**: gpt-5.1 and claude-sonnet-4.5 on Mathematical Reasoning, finding $\text{RF}^{\text{contrast}}{=}13.25\%$ vs. $86.72\%$ respectively. and we would like to explicitly frame closed-source coverage as both a scope limitation and an auditability concern.
> >
>
> Thank you for raising this point. We agree that restricting our core experiments to open-source LRMs is an important limitation, and we will make this more explicit. This choice is, however, driven by the type of intervention required for our causal notion of faithfulness.
>
> Our metric asks whether a model’s **own** stated reasoning causally drives its **own** answer under a counterfactual edit, which in turn requires modifying the reasoning trace inside the assistant turn so that the model treats the edited text as its prior thoughts rather than new user input. Current closed-API systems **do not expose this interface** (and sometimes treat the injected reasoning as **external user content** (Fig. 16)), so we cannot faithfully instantiate the same causal estimand as in Sec. 2.
>
> To still provide signal beyond open-source models, we **ran a small additional study** on the Mathematical Reasoning task with **state-of-the-art proprietary LRMs**: gpt-5.1 [1] and claude-sonnet-4.5 [2] using an adapted multi-turn protocol.
>
> |  | $\text{RF}^{\text{contrast}}$ (%) | $c(\mathcal{M})$ | $\chi(o)$ | $\chi(o')$ | $\kappa(o,o')$ |
> | --- | --- | --- | --- | --- | --- |
> | gpt-5.1 | **13.25** | 0.7629 | 0.9847 | 0.1618 | 0.9669 |
> | claude-sonnet-4.5 | **86.72** | 0.9223 | 0.9979 | 0.8736 | 0.9958 |
>
> We obtained $\text{RF}^{\text{contrast}}{=}13.25$% for gpt-5.1 (with $\chi(o') \approx 0.16$, mostly fail to identify flaw) and $86.72$% for claude-sonnet-4.5. However, since this setup necessarily differs from our single-turn, direct output-level intervention for open-source LRMs, we report these results only in the appendix and do not mix them into the main cross-model comparison.
>
> In the revision, we will clarify that this exclusion represents both (i) a limitation of RFEval’s current scope and (ii) a critical finding regarding the auditability of proprietary systems. We have included these supplementary results in Appendix G.2 of the revised manuscript.

---

> ### Author Response · Authors · 2025-11-26
>
> ### Question 1: Mismatch Between External and Own Reasoning
>
> ---
>
> > **Reviewer Concern**: How significant is the impact of the mismatch between external and own reasoning process?
> >
>
> > **Summary**: We compute baseline self-consistency and find that, **without intervention**, **Qwen3-32B and R1-Llama-70B achieve very high $\chi(o)$** (approximately **0.99** and **0.93**), indicating that their own free-form reasoning traces are already highly stance-coherent. However, **their contrast-conditional faithfulness scores are much lower** (**73.29%** and **56.47%)**, showing that $\text{RF}^{\text{contrast}}$ is mainly capturing how models behave under contradictory counterfactual reasoning, rather than trivial inconsistencies in their original explanations.
> >
>
> We appreciate the concern that externally injected counterfactual reasoning $r'$ might emphasize adherence to external rationales rather than faithfulness to the model’s own spontaneous reasoning.
>
> Conceptually, our main metric $\text{RF}^{\text{contrast}}$ is intentionally defined as a **causal** notion of reasoning faithfulness under output-level interventions: it requires both stance consistency and that the reasoning causally drives the answer. If we remove $r'$ and evaluate only the original outputs $o$, what remains is essentially $\chi(o)$, a within-output stance-coherence score that captures the plausibility/consistency of the model’s own trace but no longer tests causal influence, and thus measures a different property from the one we target.
>
> To quantify this “baseline-only” perspective, we computed $\chi(o)$ for two strong models, Qwen3-32B and DeepSeek-R1-Distill-Llama-70B, over all tasks:
>
> |  | CG | MR | LR | TR | CU | LD | PR | Overall |
> | --- | --- | --- | --- | --- | --- | --- | --- | --- |
> | Qwen-32B | 0.9561 | 0.9990 | 0.9967 | 1.000 | 1.000 | 0.9908 | 0.9669 | **0.9908** |
> | R1-Llama-70B | 0.8724 | 0.8790 | 0.9806 | 0.9800 | 0.9944 | 0.9811 | 0.6598 | **0.9261** |
>
> These numbers show that, **without any counterfactual intervention**, both models are already highly self-consistent (overall $\chi(o)\approx 0.99$ and $0.93$ respectively), although they still sometimes struggle on specific tasks (e.g., PR for R1-70B). By contrast, their contrast-conditional faithfulness scores on RFEval are substantially lower (73.29% vs. 56.47%), indicating that our main results are driven by how models **respond under contradictory reasoning interventions**, not by trivial inconsistencies in their original explanations. In other words, the “mismatch” you highlight is largely a matter of **what property is being measured**: baseline $\chi(o)$ captures **self-consistency of the model’s own reasoning**, whereas $\text{RF}^{\text{contrast}}$ probes whether the model can maintain a coherent, causal stance when confronted with plausible but flawed reasoning. We have added this distinction in the paper and report $\chi(o)$ as a complementary, baseline-only metric in Appendix F.3 of the revised paper.
>
> We sincerely appreciate the reviewer’s careful reading and constructive feedback. The issues raised here directly informed several of our revisions and, in our view, have materially improved the clarity, rigor, and impact of the manuscript.
>
> ---
>
> ### References
>
> [1] OpenAI, “*GPT-5.1 Instant and GPT-5.1 Thinking System Card Addendum*”, 2025, URL https://cdn.openai.com/pdf/4173ec8d-1229-47db-96de-06d87147e07e/5_1_system_card.pdf.
>
> [2] Anthropic, “*System Card: Claude Sonnet 4.5*”, 2025, URL https://assets.anthropic.com/m/12f214efcc2f457a/original/Claude-Sonnet-4-5-System-Card.pdf.

---

> > ### Comment · Reviewer_WEsV · 2025-11-28
> >
> > Dear Authors,
> >
> > Thanks very much for your detailed responses, which have addressed my concerns. I will keep my original positive score to show my support for this paper.

---

> > > ### Author Response · Authors · 2025-11-28
> > >
> > > Dear Reviewer WEsV,
> > >
> > > We sincerely thank you for your thoughtful feedback and important concerns. We truly appreciate the time and care you devoted to reviewing our manuscript in detail, which has greatly contributed to improving the clarity and depth of our work. We are also grateful for your support, and we deeply appreciate your engagement with our research.
> > >
> > > Thank you again for the valuable comments.
> > >
> > > Best,
> > >
> > > Authors

---

### Official Review · Reviewer_Zx74 · 2025-10-30

**Soundness:** 2
**Presentation:** 2
**Contribution:** 3
**Rating:** 4
**Confidence:** 4

**Summary:**

This paper introduces RFEval, a benchmark designed to evaluate the reasoning faithfulness of Large Reasoning Models (LRMs). It defines reasoning faithfulness based on two criteria: (1) the model maintains a coherent stance throughout its output, and (2) the reasoning causally determines the final answer. The benchmark assesses faithfulness by inserting flawed steps into the models’ reasoning traces. Experiments across 7 tasks and 12 LRMs reveal that a large proportion of intervened outputs are unfaithful, and that accuracy is only weakly correlated with faithfulness.

**Strengths:**

1.	The paper addresses reasoning faithfulness, an important yet underexplored topic. The proposed approach, probing faithfulness by performing intervention to the reasoning traces, is original.
2.	The authors conduct extensive experiments and analyze the results from multiple perspectives, providing a comprehensive evaluation.

**Weaknesses:**

1.	Overall, I am not convinced that interventions on reasoning traces truly measure model faithfulness. As the authors note, “a faithful explanation should reflect a model’s internal reasoning process.” However, modifying the output reasoning trace is not equivalent to altering the internal reasoning process and may instead confuse the model. If the model does not believe the inserted reasoning, it may fail to continue coherently, but this does not necessarily mean it would be unfaithful when reasoning autonomously.
2.	The definition of the causal influence metric lacks clarity. In Equation (3), reasoning is said to have causal influence if either the stance of the reasoning or the answer changes. However, if the reasoning stance changes but the answer stance does not (or vice versa), it should not be considered causally influential or faithful.
3.	If I understand correctly, the flaw identification in Section 4 corresponds to the measurement of stance consistency. This evaluation is conducted by LLMs, but their agreement with human judgments is not very high. This discrepancy may undermine the reliability of the reasoning faithfulness scores, especially since the analysis indicates that unfaithfulness is mainly driven by stance consistency failures.

**Questions:**

1.	What is the contrast coverage in Table 2 and what does it signify?

---

> ### Author Response · Authors · 2025-11-26
>
> Thank you for your thoughtful and detailed comments. Below, we address each of your points in turn.
>
> ### Weakness 1: Intervention on Reasoning Traces
>
> > **Reviewer Concern**: Overall, I am not convinced that interventions on reasoning traces truly measure model faithfulness. As the authors note, “a faithful explanation should reflect a model’s internal reasoning process.” However, modifying the output reasoning trace is not equivalent to altering the internal reasoning process and may instead confuse the model. If the model does not believe the inserted reasoning, it may fail to continue coherently, but this does not necessarily mean it would be unfaithful when reasoning autonomously.
> >
>
> > **Summary**: We define **unfaithfulness as a mismatch between the model’s internal decision process and its externally stated reasoning**. Our **output-level interventions are designed to expose this mismatch**, so the “confused” behaviors—where the model fails to coherently follow or explicitly correct the injected reasoning—are exactly the cases we classify as unfaithful.
> >
>
> We thank you for this fundamental question, which cuts to the core of our paper’s motivation. We agree that our output-level intervention is not *identical* to modifying the opaque internal reasoning process. In our framework, **unfaithfulness** means a mismatch between the model’s internal decision process and its externally stated reasoning, and our output-level interventions are designed precisely to surface this mismatch—so the “confused” behaviors are exactly the cases we seek to measure.
>
> Our work starts from the observation that, because the internal reasoning process is opaque (as we have noted in Sec. 2), **we must rely on a rigorous, falsifiable behavioral proxy to assess faithfulness**. Concretely, our implementation follows the “perturbing important features” paradigm [1], constructing interventions that are designed so the model’s answers are sensitive to changes in the stated reasoning.
>
> We kindly note you that the concern on **“confusion” is precisely the unfaithful behavior** we aim to detect and measure. A truly **faithful** model, when presented with a plausible (but flawed) premise $r'$, has two coherent paths:
>
> 1. **Faithful Self-Correction:** Explicitly identify the flaw in $r'$ and justify a different conclusion (captured by $\chi(o')=1$ via the $\text{Identified}(\dots)$ condition in Eq. 1).
> 2. **Faithful-Incorrect:** Fail to spot the flaw, but **coherently follow** the flawed premise to its logical (but incorrect) conclusion (captured by $\chi(o')=1$ and $\kappa=1$).
>
> The "confused" states that you posits are, in fact, **failures of stance consistency ($\neg\chi(o')$)**, and these are the **unfaithful** paths:
>
> 1. The model’s $r_\text{new}$ contradicts $r'$ without justification.
> 2. The model adopts $r'$ but its answer $a'$ ignores it, reverting to its original stance (an $r \to a$ break).
> 3. The model "silently corrects" $r'$, changing $a'$ without any justification in the reasoning trace.
>
> In all these "confused" cases, the model's output trace is **demonstrably not the true driver of its decision.** This is the very definition of a post-hoc, unfaithful justification.
>
> Therefore, our intervention is not merely "confusing" the model, and it is a valid adversarial test that successfully elicits and measures this exact failure mode, proving that the stated reasoning is not causally influential.

---

> ### Author Response · Authors · 2025-11-26
>
> ### Weakness 2: Clarification of Causal Influence (Eq. 3)
>
> ---
>
> > **Reviewer Concern**: The definition of the causal influence metric lacks clarity. In Equation (3), reasoning is said to have causal influence if either the stance of the reasoning or the answer changes. However, if the reasoning stance changes but the answer stance does not (or vice versa), it should not be considered causally influential or faithful.
> >
>
> > **Summary**: Since counterfactual reasoning $r'$ is constructed to **contradict the baseline stance**, any stance shift in $(r_{\text{new}},e',a')$ relative to $(r,e,a)$ is therefore attributed to the intervention and counted as causal influence. However, such cases still yield $\text{RF}=0$ if they break stance consistency (e.g., only reasoning or only the answer flips), so they are treated as **causally affected but unfaithful**, not as faithful explanations.
> >
>
> We appreciate this comment and fully agree that cases where only the reasoning stance or only the answer stance changes **should not** be considered faithful.
>
> A key design choice in RFEval is that the counterfactual reasoning $r'$ is always constructed to take the **opposite stance** from the model’s baseline reasoning $r$. Under this setup, we define $\kappa(o,o')=1$ whenever **any stance in the output trajectory**—reasoning or answer—differs between the baseline $(r,e,a)$ and the intervened output $(r_{\text{new}},e',a')$. Intuitively, if some stance changes while the input and model are fixed but $r'$ has been flipped, we attribute that change to the counterfactual intervention and treat it as evidence of **causal influence**.
>
> Crucially, however, it is worth noting that **reasoning faithfulness is not defined by $\kappa$ alone**. In our framework, $\kappa$ is a **necessary but not sufficient** flag that some part of the output was affected by the intervention, while faithfulness always requires the **conjunction** of causal influence and stance consistency.
>
> In the two scenarios raised by the reviewer:
>
> - **Reasoning changes, answer does not.**
>
>     Since $r'$ is constructed to oppose $r$, a change in reasoning stance without a corresponding, coherently justified link to the (unchanged) answer necessarily introduces a stance mismatch (e.g., $S(r_{\text{new}})\neq S(a')$ or $S(e')\neq S(a')$). Unless the model explicitly identifies and resolves this mismatch (captured in $\chi$), we have $\kappa=1$ but $\chi(o')=0$, hence $\text{RF}=0$: the intervention had an effect, but the resulting trace is **unfaithful**.
>
> - **Answer changes, reasoning does not.**
>
>     Symmetrically, if the answer stance flips while the reasoning stance remains aligned with the original baseline, this again creates a stance mismatch (e.g., $S(r_{\text{new}})\neq S(a')$ or $S(e')\neq S(a')$). As above, unless the model explicitly surfaces and resolves the conflict, we obtain $\kappa=1$ but $\chi(o')=0$, so $\text{RF}=0$.
>
>
> Thus, we treat these configurations exactly as the reviewer suggests: they are **not** counted as faithful, but as cases where the intervention measurably perturbs the trajectory (hence $\kappa=1$) while breaking stance consistency (hence $\chi(o’)=0$). In the revision, we will make this design explicit: (i) $\kappa$ captures *whether* the counterfactual stance intervention had any causal effect on the output, leveraging the fact that $r'$ is always stance-opposed to $r$, and (ii) faithfulness requires **both** causal influence and stance coherence, so the reviewer’s two scenarios are explicitly classified as **causally affected but unfaithful** under our full definition.

---

> ### Author Response · Authors · 2025-11-26
>
> ### Weakness 3: Reliability of Evaluation Result
>
> ---
>
> > **Reviewer Concern**: If I understand correctly, the flaw identification in Section 4 corresponds to the measurement of stance consistency. This evaluation is conducted by LLMs, but their agreement with human judgments is not very high. This discrepancy may undermine the reliability of the reasoning faithfulness scores, especially since the analysis indicates that unfaithfulness is mainly driven by stance consistency failures.
> >
>
> > **Summary**: Our stance-consistency score is a **deterministic, rule-based metric**, and the LLM evaluator only supplies stance/flaw labels rather than directly deciding faithfulness. On 1,035 component-level decisions, these labels show high agreement with humans (micro-F1 0.952 for stance, 0.938 accuracy for flaws; Cohen’s Kappa $\approx$0.92 in stance and $\approx$0.70 in flaw for both Human-Human and Human-LLM), indicating that our findings are substantially reliable.
> >
>
> We thank the reviewer for raising this concern about the reliability of the LLM-based evaluator. First, we would like to clarify that our **stance-consistency metric is defined as a deterministic, rule-based function** of a small set of discrete labels: stance assignments for $(r,e,a,r_\text{new},e',a')$ and a binary flag indicating whether the model explicitly identifies a flaw. Given these labels, $\text{RF}$ is computed **purely by fixed logical rules** (Eq. 1–5); the evaluator LLM does not directly decide faithfulness, but only **maps raw text to these discrete labels** (stance tags and identification flags).
>
> In other words, the LLM judge serves as an automatic annotator for stance extraction and flaw-identification spans, while the faithfulness score itself is fully determined by our rule-based metric. To test the reliability of this annotation step, we substantially expanded our human evaluation: across **1,035 component-level decisions** labeled by eight graduate students, the LLM evaluator achieves **0.952 micro-F1 (95% CI [0.937, 0.963]) for stance extraction** and **0.938** **accuracy (95% CI [0.922, 0.951]) for flaw identification**, indicating that the extracted labels closely match human judgments. We also compute **Cohen’s Kappa Coefficient** for both stance extraction and flaw identification, comparing Human–Human and Human–LLM agreement:
>
> |  | Human-Human | Human-LLM |
> | --- | --- | --- |
> | Stance Extraction | 0.921 | 0.938 |
> | Flaw Identification | 0.700 | 0.703 |
>
> These results support that our conclusions about stance-consistency failures are not driven by a brittle single judge, but by a deterministic metric applied to labels that are already near human level. We will clarify this pipeline (rule-based metric vs. LLM-assisted labeling) and the updated agreement statistics in the revision.

---

> > ### Author Response · Authors · 2025-11-26
> >
> > ### Question 1: Clarification of $c(\mathcal{M})$
> >
> > ---
> >
> > > **Reviewer Concern**: What is the contrast coverage in Table 2 and what does it signify?
> > >
> >
> > > **Summary**: Contrast coverage $c(\mathcal{M})$ marks **the set where our test is well-defined**, and gives a **structured handle for cross-model comparison**.
> > >
> >
> > We acknowledge that the definitions of contrast coverage may initially appear complex. The contrast coverage serves two complementary roles: **Marking the set where our causal faithfulness test is well-defined, and giving a structured handle for cross-model comparison**.
> >
> > Our contrast-conditional estimand of reasoning faithfulness measures “*How faithful is the model **when** its own baseline stance is contradicted by the injected counterfactual reasoning?*” In this setting, $c(\mathcal{M})$ tells us how often a given model even enters the regime where this causal test is applicable.
> >
> > For the detailed cross-model comparison analysis, please refer Appendix E and our 2nd response to Reviewer 4J4o. In particular, Appendix E shows that stratifying models and tasks by coverage quartiles reveals that **higher contrast coverage does not systematically inflate RF**, supporting fair cross-model comparison.
> >
> > We once again appreciate the reviewer’s thoughtful questions and conceptual clarification requests.
> >
> > ---
> >
> > ### References
> >
> > [1] Faithfulness vs. Plausibility: On the (Un)Reliability of Explanations from Large Language Models

---

### Official Review · Reviewer_JhME · 2025-10-30

**Soundness:** 2
**Presentation:** 1
**Contribution:** 2
**Rating:** 4
**Confidence:** 2

**Summary:**

The paper introduces RFEval, a new benchmark and formal framework for evaluating reasoning faithfulness in Large Reasoning Models. Faithfulness is defined via two testable criteria: stance consistency and causal influence. The authors construct 7,186 instances across seven diverse tasks and evaluate 12 open-source LRMs using output-level counterfactual interventions by injecting flawed but plausible reasoning into the model’s own thought process and observing whether the model’s answer shifts coherently.

**Strengths:**

a. The paper provides a clear, operational definition of reasoning faithfulness grounded in causal influence and logical coherence.

b. RFEval is carefully built with human-reviewed, subtly flawed counterfactual reasoning across diverse domains (math, code, law, etc.), enabling fine-grained diagnostics.

c. This study performs large-scale evaluation of 12 open-source LRMs across 7 tasks.

**Weaknesses:**

a. I personally find Section 2 hard to follow. Maybe the authors can add some examples to explain the idea.

b. Answer for Q3 in Section 5 is not convincing, as the models use different architectures and are trained on different data. Training method may not be the only factor influencing reasoning faithfulness.

c. The reasoning traces for the evaluation of faithfulness are different between models since they are filtered beforehand. The comparison is potentially not fair since they are evaluated with different question in the dataset.

**Questions:**

See weaknesses

---

> ### Author Response · Authors · 2025-11-26
>
> Thank you for your thoughtful and detailed review. Below, we address your comments point by point.
>
> ### Weakness 1: Readability of Sec. 2
>
> ---
>
> > **Reviewer Concern**: I personally find Section 2 hard to follow. Maybe the authors can add some examples to explain the idea.
> >
>
> > **Summary**: We have added **representative example** in Fig. 2 to illustrate our evaluation workflow and revised the narrative explanation between definitions in Sec. 2 to make the metric construction easier to follow.
> >
>
> We appreciate the feedback that Sec. 2 is hard to follow and we acknowledge that the assumption and definitions are presented in a somewhat raw form. We also agree that more intuition and examples would help. Intuitively, our metric $\text{RF}$ is designed to capture the idea that:
>
> *“If the is faithful for it’s reasoning, the model should coherently follow the reasoning (**Stance Consistency**) and the subsequent output should be differ when the model present different reasoning (**Causal Influence**).”*
>
> In the revision, we have add a representative example and a schematic figure (Fig. 2 in revised version) illustrating both the non-intervened (baseline) and intervened outputs, and how $\chi(o)$, $\chi(o')$, and $\kappa(o,o’)$ are computed. Also, we have revised the narrative explanation in Sec. 2 to introduce definitions in a more intuitive explanation to substantially improve readability for readers. For the another detailed example cases of evaluation, please refer the example presented in 2nd response for Reviewer Zx74.

---

> ### Author Response · Authors · 2025-11-26
>
> ### Weakness 2: Ablation of Q3
>
> ---
>
> > **Reviewer Concern**: Answer for Q3 in Section 5 is not convincing, as the models use different architectures and are trained on different data. Training method may not be the only factor influencing reasoning faithfulness.
> >
>
> > **Summary**: We address this concern with **within-family ablations,** where adding RLVR consistently lowers RF at similar coverage (MiMo: 59.33, 60.05 → **46.32**, Olmo-3: 65.87, 61.38 → **50.93**), and with a **reward analysis** showing **step rewards are actually slightly higher for stance-inconsistent** outputs (overall **0.6711** for $\chi=0$ vs **0.6280** for $\chi=1$). Accordingly, we now present post-training regime as an **important correlate**, not the sole causal driver, and treat cross-family patterns as **suggestive rather than fully deconfounded**.
> >
>
> Thank you for raising this important concern. Our intention is to seek any correlation between training paradigms and our metric to inform how future LRMs might be made more faithful. We agree that main result alone cannot isolate the effect of training method from differences in architecture and pre-training data across families. To reduce this confounding, we ran additional **within-family** ablations where these factors are approximately fixed and only the publicly available post-trained varies (MiMo-7B [1] and Olmo-3-7B [2]; Base, SFT-only, RL-only, SFT+RL).
>
> |  | MiMo ($\text{RF}^\text{contrast}$ / $c(\mathcal{M})$) | Olmo 3 ($\text{RF}^\text{contrast}$ / $c(\mathcal{M})$) |
> | --- | --- | --- |
> | Base | 59.33 / 0.69 | 65.87 / 0.42 |
> | SFT-only | 60.05 / 0.74 | 61.38 / 0.70 |
> | RL-only | **58.74** / 0.54 | - |
> | SFT+RL | **46.32** / 0.72 | **50.93** / 0.73 |
>
> Within both MiMo and Olmo 3, moving from the base model to SFT largely preserves or slightly improves $\text{RF}$, whereas adding RLVR tends to **reduce** $\text{RF}$ at comparable coverage. We interpret this as consistent with the difference in training signals: SFT, optimized via negative log-likelihood, rewards producing a fully coherent reasoning trace and answer, while current RLVR-style objectives primarily score surface format and final correctness, without explicitly encouraging stance consistency or causal influence. In the extreme, a model can obtain high reward by emitting a minimal “**reasoning shell**” (e.g., “\<think\> … Firstly … Secondly … \</think\> D: 15”) that yields the correct answer but reflects an incoherent or degenerate reasoning structure.
>
> To probe this further, we computed the **reasoning-step reward** implemented in the Open-R1 codebase [3] for the DeepSeek-R1 family, averaging over baseline and intervened outputs and stratifying by stance consistency:
>
> |  | R1-Qwen-7B | R1-Qwen-32B | R1-Llama-8B | R1-Llama-70B | Overall |
> | --- | --- | --- | --- | --- | --- |
> | $\chi=1$ | 0.6804 | **0.3996** | 0.7538 | **0.6868** | 0.6280 |
> | $\chi=0$ | **0.7200** | 0.3789 | **0.8354** | 0.6634 | **0.6711** |
>
> The average step reward is very similar for $\chi=1$ and $\chi=0$, and is in fact **slightly higher** **when outputs are stance-inconsistent**. This supports our hypothesis that existing RL objectives do not penalize (and may inadvertently tolerate) unfaithful or incoherent reasoning, even when they improve task accuracy.
>
> We will incorporate these within-family ablations and reward analyses into Sec. 5, and we will explicitly soften the claim: our results indicate that the **post-training regime is an important contributing factor** to reasoning faithfulness, but not its sole determinant. Cross-family patterns (e.g., higher RF for Qwen3-32B and LN-Super_v1 than for gpt-oss or Magistral) are presented as **suggestive, not causal**, and should be interpreted in light of remaining differences in architecture and data.

---

> ### Author Response · Authors · 2025-11-26
>
> ### Weakness 3: Conditioning & Filtering
>
> ---
>
> > **Reviewer Concern**: The reasoning traces for the evaluation of faithfulness are different between models since they are filtered beforehand. The comparison is potentially not fair since they are evaluated with different question in the dataset.
> >
>
> > **Summary**: We explicitly target a contrast-conditional estimand, evaluating $\text{RF}^{\text{contrast}}$ only on instances where the counterfactual contradicts the model’s baseline stance, since **“no-change” cases are causally ambiguous** and would confound faithfulness. Although this means different models are evaluated on different filtered subsets, we mitigate this by using **a large, diverse benchmark** and by clearly articulating the trade-off so that cross-model comparisons remain as fair and interpretable as possible.
> >
>
> We appreciate this observation and agree that our evaluation does not operate on an identical instance set across models. Conditioning on distinct stances necessarily implies that each models is evaluated on a different filtered subset. This is a direct consequence of targeting a **contrast-conditional estimand**: we define and report $\text{RF}^{\text{contrast}}$ only on instances satisfying the condition where the injected counterfactual actually contradicts the model’s baseline stance. When the stance of baseline and counterfactual reasoning share the same stance, **“no change” outcomes are causally ambiguous** (they may simply reflect the baseline stance, not the intervention), so including them would confound reasoning faithfulness with this ambiguity.
>
> To mitigate the limitation that models see different subsets of questions, we construct RFEval to be **large and diverse** across seven tasks so that the contrastive subset remains sizeable for every model.
>
> Taken together, these design choices aim to preserve **causal interpretability** of the faithfulness metric while keeping cross-model comparison as fair and transparent as possible. A closely related concern is discussed in our response to Reviewer 4J4o, and we will make this trade-off and its implications more explicit in the revision.
>
> We are grateful for these insightful comments, which have helped us clarify our methodology, sharpen our analyses, and further strengthen the paper’s contribution to understanding the reasoning faithfulness of LRMs.
>
> ---
>
> ### References
>
> [1] Xiaomi et al., “*MiMo: Unlocking the Reasoning Potential of Language Model -- From Pretraining to Posttraining*”, arXiv 2025.
>
> [2] Ettinger et al., “*Olmo 3 Technical Report*”, 2025, URL https://allenai.org/blog/olmo3.
>
> [3] Hugging Face, “*Open R1: A fully open reproduction of DeepSeek-R1*”, 2025, URL https://github.com/huggingface/open-r1.

---

### Official Review · Reviewer_4J4o · 2025-11-01

**Soundness:** 3
**Presentation:** 3
**Contribution:** 2
**Rating:** 4
**Confidence:** 4

**Summary:**

The paper proposes a formalization of reasoning faithfulness, RF (stance consistency + output‑level causal influence) and introduces RFEval, a 7,186‑instance benchmark spanning seven tasks to test whether injected counterfactual reasoning r′ coherently changes a model’s reasoning and/or answer. Evaluating 12 open‑source LRMs, the authors find
- ~50% unfaithfulness overall,
- failures concentrated in post‑intervention stance inconsistency,
- training recipe correlates with RF more than size, and
- accuracy is neither necessary nor sufficient for RF once model and task are controlled.

**Strengths:**

- Tackles a timely question of reasoning output quality and consistency
- Clear, testable behavioral definition of faithfulness separate from accuracy; simple, interpretable metrics.
- Broad multi‑task coverage and 12‑model comparison; informative diagnostics by transition location and causality type.

**Weaknesses:**

A. Granularity gap: Despite formal step‑wise notation, implementation evaluates coarse components (r/e/a), not per‑step CoT causality; this undermines a key motivation/contribution.

B. Right‑censoring: Heavy reliance on contrast‑conditional filtering (δ=1) and exclusion of truncated or malformed outputs creates informative censoring; cross‑model comparability is not fully addressed.

C. Evaluator bias/Confoundness :  A single judge with low recall on flaw identification underpins major conclusions about stance consistency.  Differences in prompts/decoding and coverage confound the hybrid vs RL‑heavy analysis. (Major Issue)

E. Weak Claim: “Accuracy is neither necessary nor sufficient for faithfulness.” Figure 5 reports weak, non‑significant association after controlling for model and task. However, the experimental setup is riddled with confounding variables, mainly LLM errors.

F. Comparison with previous work: Please contrast and cite your work against:

a. DeYoung etal., 2020 (ERASER): end‑task evaluation with deletion/insertion tests for rationales.
b. Wiegreffe & Marasović, 2021: comprehensive critique of faithfulness vs plausibility in NLP explanations.
c. Hase & Bansal, 2020 and Pruthi etal., 2020: methodology for evaluating whether rationales cause predictions under perturbations.
d. Xu et al 2025 (Re-Imagine: Symbolic Benchmark Synthesis for Reasoning Evaluation): counterfactual based reasoning benchmark.
e. On the causal side, the related literature on causal tracing / representation‑level interventions (e.g., patching internal activations) is not discussed.

**Questions:**

See weaknesses.

---

> ### Author Response · Authors · 2025-11-26
>
> Thank you for your careful and constructive review. Below, we address each of your concerns in turn. As a minor note, the figure referred to as **Figure 5** and **Table 11** in the original submission has been renumbered to **Figure 6** and **Table 13** in the revised version.
>
> ### Weakness 1: Coarse Evaluation
>
> ---
>
> > **Reviewer Concern**: Granularity gap: Despite formal step‑wise notation, implementation evaluates coarse components (r/e/a), not per‑step CoT causality; this undermines a key motivation/contribution.
> >
>
> > **Summary**: We **run a per-step variant** and observe broadly **similar task-wise patterns** (moderate positive correlation, $\rho\ge0.57$, with our metric). We therefore adopt **coarse granularity** as the primary metric, as it offers **robust and model-agnostic evaluation** across heterogeneous LRMs.
> >
>
> We appreciate the observation about the granularity gap between our step-wise formalism and the instantiated metric. We agree that step-level analysis is valuable, and our goal is not to downplay the importance of per-step CoT causality. To probe this, we ran **a pilot per-step variant of our metric** on DeepSeek-R1-Distill-Llama-8B and Qwen3-32B:
>
> - DeepSeek-R1-Distill-Llama-8B
>
> |  | CG | MR | LR | TR | CU | LD | PR | Overall |
> | --- | --- | --- | --- | --- | --- | --- | --- | --- |
> | Per-Step | 9.95 | 18.17 | 34.06 | 31.20 | 52.70 | 37.80 | 25.77 | **30.70** |
> | Ours | 26.48 | 33.03 | 55.78 | 57.68 | 64.63 | 78.97 | 94.53 | **58.46** |
> - Qwen3-32B
>
> |  | CG | MR | LR | TR | CU | LD | PR | Overall |
> | --- | --- | --- | --- | --- | --- | --- | --- | --- |
> | Per-Step | 16.61 | 35.89 | 51.67 | 65.81 | 71.09 | 51.76 | 32.37 | **47.17** |
> | Ours | 24.66 | 47.87 | 88.62 | 89.84 | 77.66 | 89.90 | 91.49 | **73.29** |
>
> As expected, the stricter per-step constraints yield lower absolute scores, but the induced **task-wise difficulty profile is broadly preserved**: across the seven tasks, the per-step and coarse metrics exhibit **moderate positive rank correlation** (Spearman $\rho\approx0.57$ for R1-Llama-8B and $\rho\approx0.68$ for Qwen3-32B), and both agree that CG/MR are the lowest-RF tasks while CU/LD lie in the high-RF regime.
>
> Despite this feasibility, we deliberately adopt the coarse $(r,e,a)$ granularity for our benchmark to maintain **robustness and model-agnosticism**. Conceptually, our formalism is intentionally defined as a step-wise level, and the $(r,e,a)$ metric should be understood not as different notion of faithfulness but as **a strict coarsening of the same causal framework**, where each block aggregates multiple micro-steps while preserving the stance and causal relations. Accurately segmenting “reasoning steps” across heterogeneous LRMs is challenging and often fragile in practice [1]: prior work typically relies on rigid, **model-specific formatting templates** [2] or **hand-crafted rule-based delimiters** [3], which do not transfer cleanly across the diverse output styles of the 12 open-source LRMs we study. Evaluating at the block level avoids these segmentation artifacts and better matches how users actually consume the reasoning trace—as a single justification leading to an answer.
>
> We will update Sec. 2 and the limitations section to explicitly state that our current implementation targets output-level, behavioral reasoning faithfulness at the $(r,e,a)$ level as a principled coarsening of our step-wise formalism. Building on the pilot per-step results reported in the rebuttal, we will also **extend the step-wise variant of the evaluation pipeline** to additional models and tasks in the camera-ready version, to further explore where step-level and block-level metrics agree or diverge within the same unified framework.

---

> ### Author Response · Authors · 2025-11-26
>
> ### Weakness 2: Conditioning ($\delta=1$), Filtering (trucated/malformed)
>
> ---
>
> > **Reviewer Concern**: Right‑censoring: Heavy reliance on contrast‑conditional filtering ($\delta=1$) and exclusion of truncated or malformed outputs creates informative censoring; cross‑model comparability is not fully addressed.
> >
>
> > **Summary**: We explicitly target a contrast-conditional estimand $\text{RF}^{\text{contrast}}$, conditioning on $\delta=1$ to **avoid causally ambiguous “no-change” cases** and treating truncated/malformed outputs as **structural failures** outside the scope of RF. To address comparability, we **co-report** contrast coverage and **analyze coverage–RF interactions**, finding that **higher coverage does not systematically inflate** $\text{RF}^{\text{contrast}}$.
> >
>
> We appreciate the reviewer’s concern about informative censoring. We would like to clarify that our design explicitly targets a **contrast-conditional estimand** rather than an unconditional average over all outputs.
>
> Concretely, we define and report $\text{RF}^{\text{contrast}}$, which is conditioned on the contrast precondition $\delta(x,r';\mathcal{M})=1$ (i.e., $S(r)\neq S(r')$). If $S(r)=S(r')$, the intervention is stance-aligned and **“no change” outcomes are causally ambiguous** (they may reflect the baseline stance rather than the injected reasoning), so including these cases in $\text{RF}$ would **confound faithfulness with this ambiguity**.
>
> We acknowledge that conditioning on $\delta=1$ implies that models are not evaluated on an identical subset of instances, which limits strict instance-matched cross-model comparability. This is, however, an inherent trade-off between causal identifiability and exact sample matching. To mitigate this, we (i) design RFEval to be **large and diverse across seven tasks** so that the contrastive subset remains sizeable for every model, (ii) **always co-report contrast coverage** $c(\mathcal{M})$ (Table 2) so that readers can see how much of the dataset each model is evaluated on, and (iii) **analyze coverage–RF interactions** in Appendix E, showing that the RF score have general trend at model- and task-level.
>
> When stratifying $\text{RF}^{\text{contrast}}$ by coverage quartiles, it peaks at mid-range coverage and **drops in the highest quartile** (by-model weighted means: 0.52, 0.57, 0.59, 0.49; by-task: 0.56, 0.53, 0.51, 0.47), and Q4 is the best-RF bin for only **2/12 models** and **1/7 tasks**, indicating that coverage does not systematically inflate RF.
>
> For malformed or truncated outputs, our intent is likewise not to hide difficult cases but to separate **structural failures** from the reasoning-faithfulness metric. As detailed in Table 11 (Appendix C.4; Table 13 in revised version), the most filtered category is “Unfinished/Truncated”. Since a repetition loop is mostly regarded as a failure mode in the study of large language models [1, 2, 3], treating them as $\text{RF}=0$ would conflate “the model failed to produce a parsable response at all” with “the model produced a coherent but unfaithful reasoning chain”, obscuring the reliability and interpretability of assessment. This careful filtering further ensures we only evaluate clearly defined scenarios, thus strengthening the validity of our measured faithfulness metrics.

---

> ### Author Response · Authors · 2025-11-26
>
> ### Weakness 3: Single Judge & Different Prompts/Decoding/Coverage
>
> ---
>
> > **Reviewer Concern**: Evaluator bias/Confoundness : A single judge with low recall on flaw identification underpins major conclusions about stance consistency. Differences in prompts/decoding and coverage confound the hybrid vs RL‑heavy analysis. (Major Issue)
> >
>
> > **Summary**: We substantially **expanded our human evaluation** (1,035 component-level decisions) and empirically find that the **LLM judge is reliable** (micro-F1 0.952 for stance, 0.938 accuracy for flaw identification; Cohen’s Kappa $\approx$0.92 in stance and $\approx$0.70 in flaw for both Human-Human and Human-LLM). Also, we reframe post-training analysis (A3) with **within-family ablations** and **reward analysis**, while carefully **limiting confounding for decoding, prompts, and coverage**.
> >
>
> We acknowledge that previous human judgements show relatively low agreement with the LLM evaluator. We have substantially **expanded our human evaluation** and now provide aggregate evidence that the LLM judge is both reliable and not acting as a brittle single point of failure. Specifically, we extended the original study to a **total of 1,035 annotated component-level decisions**, where eight graduate students independently labeled stance and flaw identification on randomly sampled instances from all tasks. On this enlarged sample, the LLM evaluator achieves an overall **micro-F1 of 0.952 (95% CI [0.937, 0.963]) for stance extraction** and an overall **accuracy of 0.938** **accuracy (95% CI [0.922, 0.951]) for flaw identification**. We also compute **Cohen’s Kappa Coefficient** for both stance extraction and flaw identification, comparing Human–Human and Human–LLM agreement. The resulting coefficients are **0.921 (stance)** and **0.700 (flaw)** for Human–Human, and **0.921 (stance)** and **0.703 (flaw)** for Human–LLM, showing **substantial agreement** between human and LLM.
>
> These results directly support the robustness of our conclusions about stance consistency, and the 87.5% overall accuracy on flaw identification further indicates that the single evaluator rarely exhibits the “low recall” failure mode. We will clarify these updated human–LLM agreement statistics in the revision to better convey the reliability of the judge.
>
> We also agree that our comparison between hybrid and RL-heavy pipelines cannot be fully controlled by inspecting our main results alone. As detailed in our 2nd response to Reviewer JhME, we therefore reframe the training-paradigm analysis with **within-family ablations** (MiMo-7B, Olmo-3-7B) and a **reward analysis** on DeepSeek-R1 family, which jointly suggest that **adding RLVR can reduce RF** (RF **$\approx$ 14%p drop** in MiMo, **$\approx$10%p drop** in Olmo 3) and that **current rewards do not penalize stance-inconsistent traces** (overall reward **0.6711** for $\chi=0$ vs. **0.6280** for $\chi=1$)**.** Accordingly, we present the post-training regime as an important correlate of RF rather than a fully deconfounded causal factor, while **taking several steps to carefully limit confounding**:
>
> 1. First, all models are decoded with the **same greedy setting** (temperature = 0), so decoding differences do not explain the observed gaps.
> 2. Second, apart from model-specific tagging (e.g., system/assistant tokens, <think> markers), the **core task instructions and reasoning prompts are shared** across models (Appendix C.2).
> 3. Third, we explicitly co-report contrast coverage $c(\mathcal{M})$ and, as shown in Appendix E, stratifying models and tasks by coverage quartiles reveals that RF typically peaks at mid-range coverage and often drops in the highest quartile, indicating that **higher coverage does not systematically inflate RF**.

---

> ### Author Response · Authors · 2025-11-26
>
> ### Weakness 4: Weak Claim on A4
>
> ---
>
> > **Reviewer Concern**: Weak Claim: “Accuracy is neither necessary nor sufficient for faithfulness.” Figure 5 reports weak, non‑significant association after controlling for model and task. However, the experimental setup is riddled with confounding variables, mainly LLM errors.
> >
>
> > **Summary**: We clarify that **accuracy and RF are conceptually distinct**, and Fig. 6 serves as **empirical evidence** that the accuracy–RF association is small and statistically insignificant. Moreover, only RF depends on the LLM judge (since accuracy uses ground truth) and our expanded human study shows the judge is reliable, so **remaining evaluator noise would not attenuate** the already weak correlation.
> >
>
> We agree that Fig. 5 (Fig. 6 in revised version) alone should not be over-interpreted as a strong causal statement. Our claim that “accuracy is neither necessary nor sufficient for faithfulness” is, however, not based solely on the regression in Fig. 5. Conceptually, our definitions already admit **faithful-incorrect** cases (coherent stance and causal influence leading to an incorrect answer) and **unfaithful-correct** cases (e.g., silent corrections where the answer is correct but the reasoning–answer chain violates $\chi$), so accuracy and faithfulness are not logically coupled.
>
> Fig. 5 is intended as a **supporting, not primary** evidence. After controlling for systematic differences across models and tasks, the residual association between accuracy and RF is small and statistically insignificant, indicating that ***accuracy is not a reliable proxy for RF*** in our setting. Regarding potential “LLM errors,” only RF depends on the LLM-based evaluator, whereas accuracy is computed directly against ground truth. We would like to kindly note that our expanded human evaluation (1,035 component-level decisions) shows that the evaluator achieves 0.952 micro-F1 for stance extraction and 0.938 accuracy for flaw identification; Cohen’s Kappa $\approx$0.92 in stance and $\approx$0.70 in flaw for both Human-Human and Human-LLM, so remaining evaluator noise primarily **attenuates** correlations, making our estimate of the accuracy–RF link conservative.
>
> We will further clarify this in the revision and tone our wording to emphasize that our conclusion concerns the lack of a strong, predictive relationship, not a stronger causal claim.

---

> ### Author Response · Authors · 2025-11-26
>
> ### Weakness 5: Related Works Comparison
>
> ---
>
> > **Reviewer Concern**: Comparison with previous work.
> >
>
> > **Summary**: We have **updated and added related-work paragraph** in the revised paper.
> >
>
> We thank you for pointing out these relevant lines of work and will clarify our positioning with respect to them.
>
> ERASER [4], Hase & Bansal [5], and Pruthi et al. [6] primarily study whether input rationales affect predictions, whereas ours operates on **output-level counterfactual interventions** to test whether the model’s own generated reasoning causally governs its answer under stance-level perturbations.
>
> Wiegreffe & Marasović [7] focuses on the taxonomy and collection of human-annotated textual explanations and provides guidelines for building explanation datasets. Our work shifts the emphasis from dataset design to the model’s own reasoning process by formalizing $\text{RF}$, and instantiate this as an output-level counterfactual intervention.
>
> Xu et al. [8] synthesize symbolic counterfactual benchmarks for reasoning inputs*.* While informative, our counterfactuals target the **reasoning trace itself** rather than mutating input and are evaluated across 12 LRMs and seven heterogeneous tasks.
>
> Finally, causal tracing and representation-level interventions (e.g., activation patching) aim to localize internal causal pathways in hidden states [9, 10], which is complementary but orthogonal to our **behavioral** notion of reasoning faithfulness. We do not claim to identify mechanistic circuits, but instead provide an end-to-end, model-agnostic protocol for auditing whether the textual reasoning a user sees is structurally aligned with and causally determinative of the final answer.
>
> We have added a detailed related-work paragraph to make these distinctions explicit and cite all of the above of the revised paper.
>
> We are grateful for these insightful comments, which have encouraged us to sharpen our exposition, deepen the analyses, and further improve the overall clarity and contribution of the paper.
>
> ---
>
> ### References
>
> [1] Liu et al., “*AdaptiveStep: Automatically Dividing Reasoning Step through Model Confidence*”, ICML 2025.
>
> [2] Chen et al., “*Unveiling Chain of Step Reasoning for Vision-Language Models with Fine-grained Rewards*”, NeurIPS 2025.
>
> [3] Lee et al., “*Token-Supervised Value Models for Enhancing Mathematical Problem-Solving Capabilities of Large Language Models*”, ICLR 2025.
>
> [4] DeYoung et al., *“ERASER: A benchmark to evaluate rationalized NLP models”*, ACL 2020.
>
> [5] Hase & Bansal, *“Evaluating explainable AI: Which algorithmic explanations help users predict model behavior?”*, ACL 2020.
>
> [6] Pruthi et al., *“Estimating Training Data Influence by Tracing Gradient Descent”*, NeurIPS 2020.
>
> [7] Wiegreffe & Marasović, *“Teach Me to Explain: A Review of Datasets for Explainable Natural Language Processing”*, NeurIPS 2021.
>
> [8] Xu et al., *“RE-IMAGINE: Symbolic Benchmark Synthesis for Reasoning Evaluation”*, ICLR 2025 Workshop.
>
> [9] Zhang & Nanda, *“Towards best practices of activation patching in language models: Metrics and methods”*. ICLR 2024.
>
> [10] Che et al., *“Separating Tongue from Thought: Activation Patching Reveals Language-Agnostic Concept Representations in Transformers”*, ACL 2025.

---

### Author Response · Authors · 2025-12-03
**Final Remarks by Authors**

We sincerely thank the Area Chair for overseeing the review process and all reviewers for their careful, constructive feedback on our study of reasoning faithfulness in Large Reasoning Models (LRMs).

We are grateful that several reviews highlighted the **strengths and value** of our work, including:

- Addressing a **timely, important, and under-explored topic** (Reviewers 4J4o, Zx74),
- Providing a **clear, formal, and testable definition of reasoning faithfulness** via **output-level interventions** on the reasoning trace (Reviewers 4J4o, JhME, WEsV),
- Conducting a **large-scale evaluation** of 12 open-source LRMs across **diverse domains** (all reviewers).

Reviewers also raised **important concerns** regarding (1) conditioning and filtering, (2) reliability of the LLM evaluator, (3) cross-model comparison, and (4) the validity and interpretation of our metric. During the discussion, we addressed these points with additional analyses and clarifications:

- **Validity and interpretation of the metric** (1st response to Reviewer 4J4o, 2nd response to Reviewer Zx74, 1st response to Reviewer WEsV): We explain why the implemented block-level $(r,e,a)$ granularity should be understood as a strict coarsening of our step-wise formalism, chosen for **robustness and model-agnostic evaluation** across heterogeneous LRMs. We clarify that reasoning faithfulness is defined by the **conjunction of causal influence and stance consistency**, so the traces that do not drive the answer under intervention are explicitly classified as unfaithful.
- **Conditioning and filtering** (2nd response to Reviewer 4J4o, 3rd response to Reviewer JhME): We explicitly target an $\text{RF}^\text{contrast}$  to **avoid causally ambiguous “no-change” cases** and treat truncated/malformed outputs as **structural failures** outside the scope of RF, avoiding a conflation of parsing failures with genuine unfaithful reasoning. We co-report contrast coverage and analyze coverage–RF interactions to show that higher coverage does not systematically inflate RF.
- **Reliability of the LLM evaluator** (3rd response to Reviewer 4J4o, 3rd response to Reviewer Zx74): We clarify that stance consistency is computed by a deterministic rule-based metric over discrete labels (stance tags and flaw-identification flags); the **LLM evaluator only maps text to these labels**. We expanded our human evaluation to 1,035 component-level decisions and show that the **LLM judge achieves high agreement** with humans (micro-F1 and accuracy, with Cohen’s Kappa comparable to human–human agreement), indicating that the extracted labels are stable and the metric is not driven by a brittle single judge.
- **Cross-model comparison and training paradigm** (2nd response to Reviewer JhME, 2nd response to Reviewer WEsV): We reframe our analysis as observational rather than causal and reduce confounding from architecture and pre-training differences via **within-family ablations** (MiMo-7B, Olmo-3-7B) and a **reasoning-step reward analysis** on the DeepSeek-R1 family. These indicate that RL-style post-training (RLVR) can reduce RF and that current rewards do not penalize stance-inconsistent traces, so we present post-training regime as an important correlate of RF, while treating cross-family patterns as suggestive rather than fully deconfounded.

In addition, we explicitly discuss the **limited scope to open-source LRMs** as a real limitation of RFEval, and report a **small pilot study on state-of-the-art closed-API models** (gpt-5.1, claude-sonnet-4.5) under an adapted multi-turn protocol in the appendix, clarifying why current APIs do not allow us to faithfully instantiate the same causal estimand.

Reviewer WEsV has indicated that their concerns were addressed and maintained their scores after our responses. While Reviewers 4J4o, JhME, and Zx74 have not yet posted follow-ups, we believe the revised manuscript and the added analyses directly resolve the main points of uncertainty and clarify the intended scope and interpretation of our claims.

In summary, we believe the revised submission now presents a clearer and more rigorous evaluation of reasoning faithfulness in LRMs—both in terms of formal definitions and empirical analysis—and we are deeply grateful to the Area Chair and reviewers for the feedback that led to these improvements.

Best regards,

Authors

---

### Meta-Review · Area_Chair_McMP · 2026-01-10

**Summary:**

Given the nature of this submission as a benchmarking approach, the concerns of the reviewers span various topics:
* Presentation: This is important because benchmarking requires precise and convincing terminology of what is measured, how and why these choices were made
* How convincing are the methodologies to link measurements to conclusions? For example Q3 sec 5, or how convincingly the interventions truly measure faithfulness.
* Experimental results and ablations, tackling mismatch in datasets and models. A related concern is around the granularity being too coarse as well as potential bias in the evaluation.

**Reviewer Concerns:**

The authors presented a detailed rebuttal with new experiments and clarifications.

__Presentation:__ I find that the presentation issues are largely resolved given the discussion and the authors’ clarifications around what is (and is not) claimed.

__Supporting conclusions:__
This is substantially (though not perfectly) addressed. The answer to Q3 in Sec 5 is strengthened with the new ablations, and the authors convincingly separate what is intended to be a behavioral notion of reasoning faithfulness from stronger claims about internal mechanistic circuits. I still think that the operational notion of “faithfulness” invites some reasonable disagreement, but at this point the remaining ambiguity is more a scope issue rather than a blocker for publication. Importantly, as a benchmarking paper, the bar is that the formulation is tight enough to support consistent use and meaningful comparisons, and with the added clarifications the paper gets much closer to that bar.

__Experimental results and ablations, tackling mismatch in datasets and models and bias issues.__
The additional analyses improve the credibility of the empirical story, especially around evaluation reliability and confounding. Reviewer 4J4o’s concern on evaluator bias is addressed more convincingly with the expanded human evaluation. There could be scope for further elimination of bias concerns through more insightful cross-judge analyses, but I find that the additional analyses provide a convincing enough picture.

Regarding the granularity concern, this is resolved to an extent with new helpful experiments. As the authors note, this more granular set-up is more tricky. In a sense, the concern partially remains that the focus on the coarse granularity is limiting in terms of usefulness, but for the scope of the paper the credibility is there.

**Reviewer Scores:**

Given the addressing of several concerns as discussed above, I would expect a moderate raise in scores by the three reviewers who gave a score of "4" (the reviewer who gave a score of "6" has explicitly mentioned that they keep their score). Overall, since all of the scores started close to the borderline before the rebuttal (i.e. no reviewer was feeling strongly against this paper) and the rebuttal has been helpful in shaping this paper a solid contribution, I think that with a full discussion period the overall score would move closer to “weak accept territory”.

---

### Decision · Program_Chairs · 2026-01-26

Accept (Poster)